# Dimension Reduction and Analysis of a 10-Year Physicochemical and Biological Water Database Applied to Water Resources Intended for Human Consumption in the Provence-Alpes-Côte d'Azur Region, France

**Abdessamad Tiouiouine [1], Suzanne Yameogo [2], Vincent Valles [3], Laurent Barbiero [4,*], Fabrice Dassonville [5], Marc Moulin [6], Tarik Bouramtane [1], Tarik Bahaj [1], Moad Morarech [7] and Ilias Kacimi [1]**

[1] Geoscience, Water and Environment Laboratory, Faculty of Sciences, Mohammed V University, Avenue Ibn Batouta, 10100 Rabat, Morocco; touiouineabdessamad@yahoo.fr (A.T.); tarik.bouram@gmail.com (T.B.); tarikbahaj@yahoo.fr (T.B.); iliaskacimi@yahoo.fr (I.K.)

[2] Earth Sciences Department, Université de Ouagadougou (Pr J. Ki-Zerbo), B.P. 7021 Ouagadougou 03, Burkina Faso; suzanneyameogo@yahoo.fr

[3] UMR 1114 INRAE EMMAH Avignon Université, 84916 Avignon, France; vincent.valles@univ-avignon.fr

[4] IRD, CNRS, Université de Toulouse, UMR 5563, Géoscience Environnement Toulouse, Observatoire Midi-Pyrénées, 14 Avenue Edouard Belin, 31400 Toulouse, France

[5] ARS Provence-Alpes-Côte d'Azur, 132, Boulevard de Paris, 13002 Marseille, France; Fabrice.DASSONVILLE@ars.sante.fr

[6] BRGM, Direction régionale Provence-Alpes-Côte d'Azur, 117 avenue de Luminy, 13009 Marseille, France; m.moulin@brgm.fr

[7] Laboratoire en Géosciences appliqués et marines, Géotechnique et Géo risques (LR3G), Essadi-Faculté des Sciences, Université Abdelmalek, 93000 Tétouan, Morocco; Morarech2000@gmail.com

\* Correspondence: laurent.barbiero@get.omp.eu

**Abstract:** The SISE-Eaux database of water intended for human consumption, archived by the French Regional Health Agency (ARS) since 1990, is a rich source of information. However, more or less regular monitoring over almost 30 years and the multiplication of parameters lead to a sparse matrix (observations × parameters) and a large dimension of the hyperspace of data. These characteristics make it difficult to exploit this database for a synthetic mapping of water quality, and to identify of the processes responsible for its diversity in a complex geological context and anthropized environment. A 10-year period (2006–2016) was selected from the Provence-Alpes- Côte d'Azur region database (PACA, southeastern France). We extracted 5,295 water samples, each with 15 parameters. A treatment by principal component analysis (PCA) followed with orthomax rotation allows for identifying and ranking six principal components (PCs) totaling 75% of the initial information. The association of the parameters with the principal components, and the regional distribution of the PCs make it possible to identify water-rock interactions, bacteriological contamination, redox processes and arsenic occurrence as the main sources of variability. However, the results also highlight a decrease of useful information, a constraint linked to the vast size and diversity of the study area. The development of a relevant tool for the protecting and managing of water resources will require identifying of subsets based on functional landscape units or the grouping of groundwater bodies.

**Keywords:** hydrochemistry; water resource; hydrogeology; multivariate statistics; France

## 1. Introduction

Groundwater conveys information acquired during its journey from the surface to the aquifer [1]. Its composition reflects the inputs from the atmosphere, and the interactions with the vegetation cover, presence of agricultural or industrial activities, characteristics of the soil through which it percolated, and lithology of the geological formations that shelter the aquifers [2–9]. Thus, the chemical and bacteriological composition of the water is a source of information that can help characterize different landscape units and the impact of human and industrial activities [10].

The chemical and biological composition of groundwater is a measure of its suitability as a source of water for human and animal consumption, irrigation, and for industrial and other purposes. In France, health monitoring of the water quality intended for human consumption is carried out by Health Regional Agencies (Agence Régionale de la Santé or ARS). Over 32,000 catchments are monitored, distributed as 96% and 4% of groundwater and surface water, respectively. The results of this monitoring are archived in a database named SISE-Eaux, containing data of various nature, namely physico-chemistry, composition in major ions, microbiological parameters of fecal or non-fecal origin, heavy metals, etc. This water quality database has been continuously supplied since 1990 and has been in a digitized format for computers for more than 15 years.

The notion of water quality is complex, and presenting it in a synthetic way to access a typology and a relevant mapping of water resources can be difficult for various reasons [11,12]. The development of analytical tools has led to more precision, lower detection limits, but especially to the multiplication of the measured parameters, but resulting in information distributed in a hyperspace with many dimensions [13]. In the SISE-Eaux database, the number of parameters can reach more than one hundred for some water samples. The mapping of each parameters quickly leads to a methodological impasse because of their sheer complexity. As a result, the use for water quality mapping can be complicated, cumbersome and inevitably lead to many redundancies between parameters. The use of mathematical methods synthesizing information becomes indispensable [14–16].

This work is based on classical statistical methods derived from linear algebra and aims to reduce the dimensionality of the SISE-Eaux database to identify, discriminate, survey large hydrochemical ensembles, and link them with landscape units. A method that concentrates information to reduce and simplify the treatment of a large dataset, and allow identifying and synthetic mapping of the main sources of variability, is applied to the Provence Alpes Côte d'Azur region (PACA) in southeastern France, a region with complex and varied lithology, associated with a significant altitudinal gradient and demographic pressure with a strong seasonal character.

## 2. Materials and Methods

### 2.1. The Provence-Alpes-Côte d'Azur Region

The Provence-Alpes-Côte d'Azur (PACA) region constitutes the southeastern part of the French metropolitan territory. From an administrative point of view, the PACA region comprises 6 departments, namely the Alpes-de-Haute-Provence, Hautes-Alpes, Alpes-Maritimes, Bouches-du-Rhône, Var and Vaucluse. With an area of 31,400 km$^2$, the PACA region spans a wide altitudinal range, from the coast to the peaks of the Hautes-Alpes (Figure 1). The western end consists of the Camargue, an alluvial plain of the Rhone Delta at very low altitude. In the South and Southern region, the rocky coast resulting from the Pyreneo-Provençal folding consists of high limestone massifs plunging into the sea. In addition there are two ancient coastal massifs made up of crystalline rocks, the Maures (780 m), Esterel (618 m) and Tanneron (513 m) massifs. The Alpine foothills, resulting from the Alpine fold, form a transition zone before the high peaks of the alpine arc. Many peaks of the Pre-Alps range from 500 to 1900 m above sea level, while the high summits to the northeast of the region can exceed 4000 m (Barre des Ecrins 4102 m).

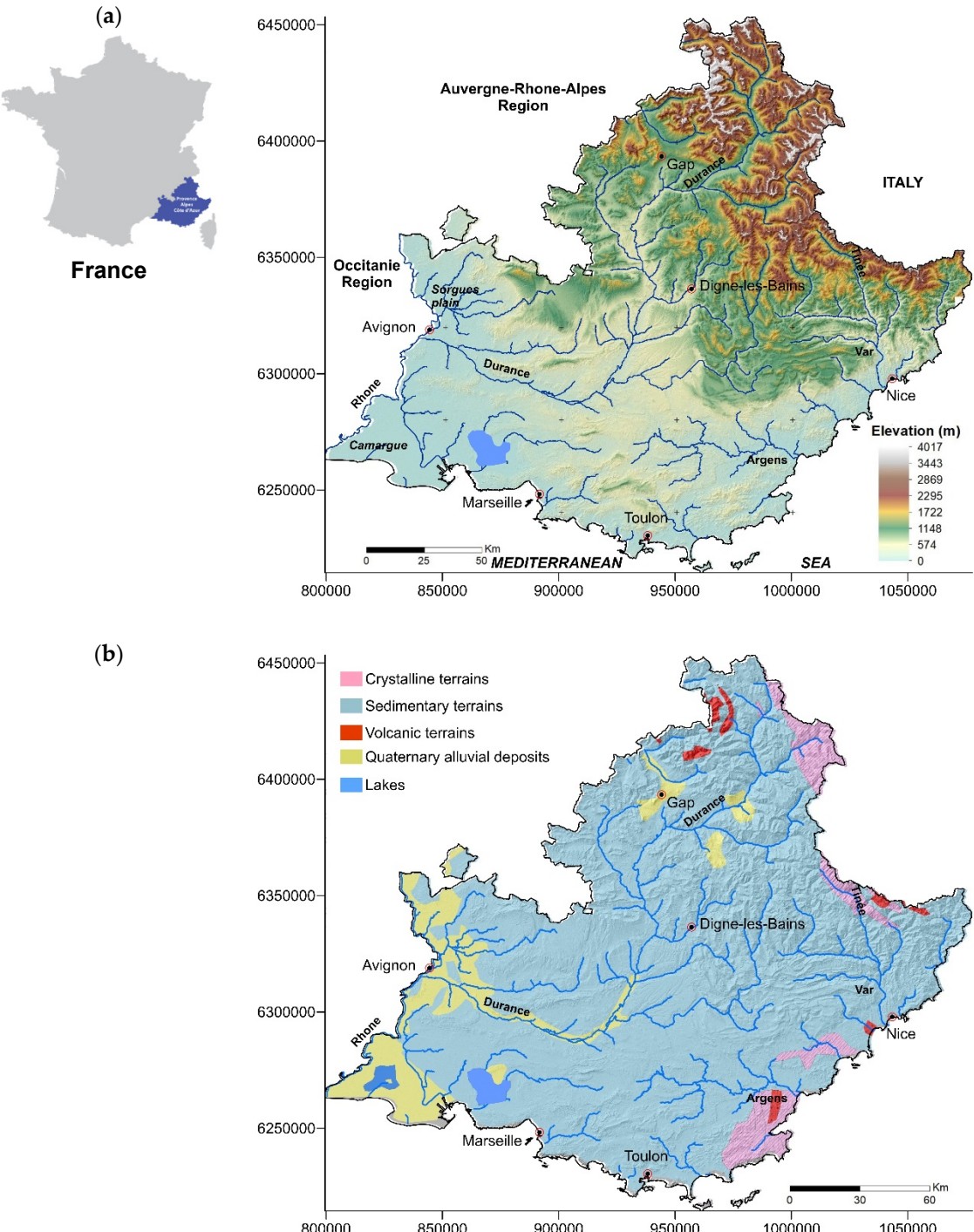

**Figure 1.** Location of the PACA region, elevation (**a**) and main rock types (**b**) of the PACA Region. Coordinates are lambert 93.

The geological context of the PACA region (Figure 1) has been the subject of many studies since the 19th century. In the Palaeozoic, the uplift of the southern part of the Hercynian chain is followed by an intense erosion causing the filling of the Permian depression surrounding the crystalline massifs (Maures and Tanneron). During the same period, volcanic activity discharged large quantities of lava at the origin of ryolites in the Esterel massif. In the secondary era, marine transgressions deposited large layers of Triassic sediments, namely sandstone, dolomitic limestones and marls, then evaporites and clays during the regression at the end of the Triassic. The sea returns in the Jurassic and Lower

Cretaceous period, depositing marls, dolomitic limestones and reefs. These deposits will then lift, slide, and fold during the formation of the Pyrenees and the Alps [17], and form the alpine chains of Provence, namely Luberon, Alpilles, Arc de Castellane, and reliefs of Sainte Baume, Sainte Victoire, Etoile, Nerthe). In the Cenozoic, Provence emerges and the reliefs erode, giving continental detrital deposits [18]. The Rhone corridor collapses, the Pyrenean chain of Provence collapses under the sea and the Gulf of Lion opens [19]. The great diversity of terrains can be divided into three main types:

- Crystalline terrains, located mainly in the Maures, and Tanneron coastal massifs and in the Alps (north to northeast of the region, particularly the Mercantour Massif), including granites, gneisses, micaschists, phyllades, diorites.
- Sedimentary, marine or lacustrine terrains, of very varied natures (clays, marls, shales, marly-limestones, limestones, dolomites, grauwackes, conglomerates, sandstones, sands, molasses, pyritous marls, lignites, and old and recent alluvial deposits of loamy, clayey, sandy or gravelly texture with pebbles).
- Volcanic terrains, mainly represented in the coastal Esterel massif, with basalts, rhyolites, and volcanic ashes.

Such a wide lithological variety is accompanied by very different water-rock interactions distributed in space as follows: In its northeastern part, it is dominated by facies alternating between gneiss, granite, shale-sandstone, ophiolite. Pre-Alps are essentially covered with a calcareous-pyritous-marls complex. The central part of the region consists of calcareous-marls-gypsum alternation and sandstone. In the southwest region, the calcareous marls lithological facies prevails over the three departments, associated in Vaucluse with sand, sandstones and clays. These different terrains, which appear in masses or in layers exhibit rapid variations of nature (vertically and often laterally). They have been affected by very large tectonic pressure from the south, resulting in variable dipping, thrust faults and overlaps of widely varying scales, shear zones, and highly developed fractures, hence, frequent sensitivity to earthquakes.

From the hydrogeological point of view, the PACA aquifers are relatively fragmented with variable dimensions because of the rapid lateral and vertical variations in the nature and therefore in the permeability of the material. Their structural conditions generate a diversity of behaviors with regard to climatic hazards and anthropic pressures [20,21]. There are five main types of aquifers: (1) fissured bedrock aquifers within crystalline massifs, (2) karst aquifers with fluctuating flow in close interaction with rainfall [22]. These aquifers have great potential, which is generally underexploited; (3) Complex aquifers of the Alpine domain; (4) Complex aquifers of the Provençal domain and (5) coarse alluvial aquifers, which are the most exploited due to the proximity of the resource and strong human activity along the hydrographic network. Near the coast, these aquifers are subject to seawater saline intrusions (Figure 2).

The Rhone dominates the hydrographic network in the PACA region to the west and forms the limit with the Occitanie region (Figure 1). The plain of Camargue occupies its wide delta. The Durance (320 km) is the main tributary of the Rhone, with a watershed covering 45% of PACA. This quasi-torrential river is the main axis of penetration in the northeastern mountainous part of the region. Relatively small coastal rivers irrigate the Côte d'Azur, namely the Argens (116 km) and Var (120 km).

To this geological and altitudinal diversity is added a diversity in the land use together with in addition a marked seasonal component, in particular in the migration of the herds towards the altitude and the influx of tourists towards the coastal areas in summer.

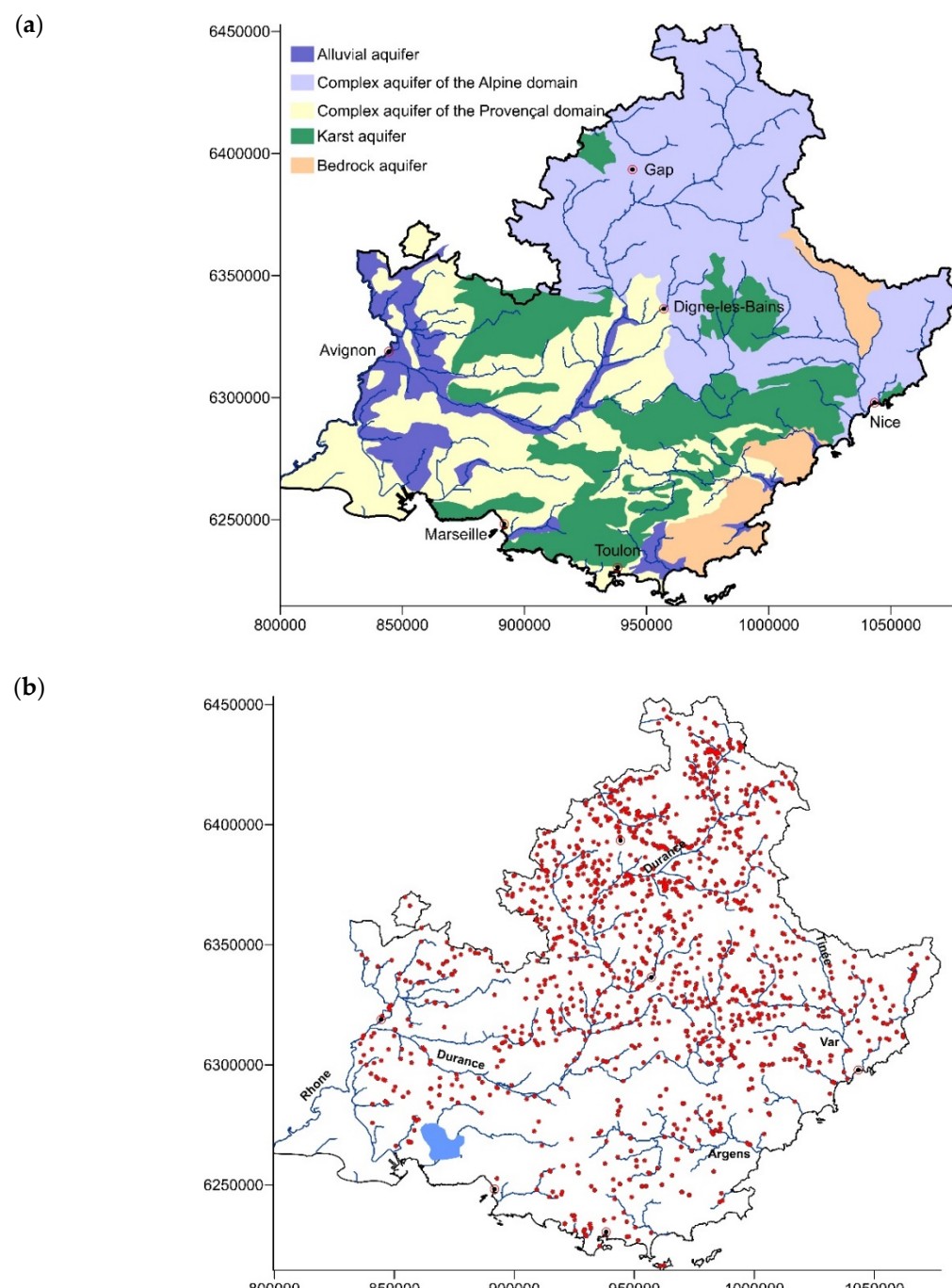

**Figure 2.** Main aquifer types (**a**) and location of the 1061 groundwater sampling points (**b**).

## 2.2. Water Database

The water samples were collected and analyzed by the ARS—PACA service provider laboratories, approved by the Ministry of Health, and having all the international certifications for analytical quality. The dataset used in this work was extracted in two stages from the ARS—PACA database. A first extraction concerned raw water analysis, i.e., before any treatment. More than 10 thousand water samples with up to 30 microbiological and/or physicochemical parameters were obtained. The data are spread over a 10-year period between 10 January 2006 and 15 September 2016. Three different types of analyses were identified, i.e., complete analyses grouping many parameters (up to 100), standard analyses comprising only a few parameters (about 30) and control analyses generally focused on a limited number of parameters (about 10). As a result, the matrix obtained from this first extraction

was a sparse matrix. The second step consisted of eliminating empty cells. This results in a block of 5,295 analyses and 15 parameters, namely the biological parameters Enterococci (Enteroc.) and *Escherichia coli* (E. coli), the electrical conductivity (EC), major ions referred to as $HCO_3$, Cl, $SO_4$, $NO_3$, $H^+$, Ca, Mg, K, Na, metals Fe, Mn, and metalloid As, constituting a full matrix of 79,425 values. The pH of the water was transformed into $H^+$ ion concentration to avoid mixing the parameters with linear or logarithmic scale and thus have a homogeneous dataset. As the samples were not all georeferenced, the missing coordinates were acquired from the sites Infoterre (infoterre.brgm.fr), Géoportail (www.geoportail.gouv.fr), and/or from the ADES database (ades.eaufrance.fr). For the health monitoring, the water resources supplying the big cities were collected on the same point at different dates, sometimes providing up to several tens of samples, which allowed for good repeatability, and a temporal representativeness.

*2.3. Data Treatment*

A principal component analysis (PCA) was carried out by diagonalization of the correlation matrix in order to reduce the volume of redundant information, to identify, quantify and rank the different independent sources of variability, and to explore the underlying processes responsible for the variability in water quality. PCA is a very efficient and widely used method for studying a mass of data with many parameters. On the one hand, several processes or sources of variability can affect the parameters (or variables) with a different intensity. For example, the calcium content can be affected by dissolution or precipitation of calcite, ion exchange, or the supply of liming product, etc. On the other hand, each process can influence several parameters with different intensities. Thus, for example, the redox potential can influence the nitrate contents (through denitrification), manganese and dissolved iron. The PCA treatment makes it possible to discern the relationships between the different parameters and to rank the independent sources of variability of the information contained in the database. The procedure, which uses the correlation matrix, includes mean-centering variables that sidestepped problems arising from the variable numerical ranges and units by automatically autoscaling all variables to the mean zero and variance unit. It transforms n original variables into n orthogonal principal components that are a linear combination of the original variables. Orthogonal (uncorrelated) eigenvectors related to the principal components ensure the independence of the associated processes. The PCA was followed by an orthomax normalized rotation, making it easier to interpret the contributions of the principal components, while preserving the independence of the sources of variability. The data dimension reduction performed by the PCA was assessed using the Bartlett sphericity test. The calculations were done using the XL-Stat (Addinsoft, Bordeaux, France) extension of Microsoft Excel.

*2.4. Mapping*

Principal components furnish macro parameters, i.e., synthetic data that convey strong and significant information and, therefore, are particularly suitable and relevant for time monitoring or digital mapping and spatial analysis, just like original parameters [14,23–25]. The scores of the samples along the principal components were used for the mapping of the main sources of variation in water quality identified in the study. Maps were created using adjusted variograms for the Kriging (SURFER 12; Golden Software®, Golden, Colorado, USA).

## 3. Results

The location of the 1,061 sampling points on the PACA region is shown in Figure 2. The large number of data ensured a good reliability of calculations. The entire region was covered, although some areas had lower sampling densities. This can be explained either by the scarcity of water as in the case of karst areas with a very deep water table, or by the presence of very superficial water and being of too poor quality to be suitable for drinking water supply, as for example in the Rhone delta.

The alpine and mountain areas showed many sampling points because of dispersed habitat between the different valleys.

## 3.1. Principal Component Analysis

The correlation matrix (Table 1) showed generally poor correlation coefficients. Only the Electrical Conductivity (EC) was significantly correlated with soluble major elements ($Ca^{2+}$, $SO_4^{2-}$, $Mg^{2+}$, $Cl^-$, $Na^+$) with correlation coefficients close to 0.75–0.85. These major ions were moderately correlated with each other, with the correlation between $Na^+$ and $Cl^-$ being the highest. Apart from this group of parameters, the correlation coefficients were low.

The first 6 factorial axes accounted for 75.7% of the information, and $PC_1$ to $PC_4$ have eigenvalues greater than 1 (Figure 3 and Table 2), meaning that they concentrate more information than one single variable [14]. The first principal component stood out clearly from the other axes. It alone accounted for 34.1% of the total variance, that is, more than a third of the information contained in the database. It was positively scored with EC and the major ions (Ca, Mg, Na, Cl, $SO_4$, $HCO_3$, K) and can be considered an axis of water dissolved load. The second PC (11% of the variance) was positively correlated with the bacteriological parameters, together with the variables $HCO_3$, Ca, and $NO_3$, and negatively with Na and Cl. It is therefore an axis reflecting the bacterial contamination in carbonate and karstic aquifers. The first factorial plane ($PC_1$ and $PC_2$) accounted for 45% of the total variance, any sample and any parameter taken together, that is to say that the analysis of the $PC_1$–$PC_2$ score plot makes it possible to visualize almost half of the total information (Figure 3). The third PC ($PC_3$, 10.4% of the variance) was also positively scored with the bacteriological parameters *Enterococci* and *Escherichia coli*, but also with Na and Cl, and negatively with $HCO_3$, Ca, Mg, $NO_3$. It can be regarded as conveying the contamination of aquifers with Na-Cl dominated chemical profile. The fourth PC (7.3% of the variance) discriminated Fe and Mn-laden waters with lower pH. $PC_5$ (6.7% of the variance) reflected the presence of arsenic, a toxic and carcinogenic metalloid usually detected in low concentration. Finally, $PC_6$ (6.3% of the variance) was positively correlated with the $H^+$ (low pH values) and $NO_3$, and negatively with Fe and Mn.

It is difficult to distinguish an associated process for PCs that account for a low percentage of the variance. Therefore, $PC_7$ and above were not considered in the study due to their low eigenvalues, assuming that they make up part of the chemical background noise of the sample set [14,26]. The Bartlett sphericity test showed calculated $\chi2$ (=70371), greater than the critical $\chi2$ (=130, significant level 0.05 and p-value <0.0001). Thus, the shift from 15 parameters to 6 principal components, i.e., a 6-dimensional hyperspace of data, constitutes a strong dimensional reduction, although it only resulted in a loss of 24.3% of the information. Such a hyperspace reduction facilitates the information processing and classification. The statistical noise, inherent to any database is included in these 24.3%.

After orthomax rotation, the three first principal components are referred to as $D_1$, $D_2$, and $D_3$ (Figure 4). $D_1$ was clearly associated with the carbonate calcium and magnesian chemical profile, and was positively scored with EC, i.e., scored according to the water mineral load. $D_2$ was, in turn, correlated with Na and Cl, also positively scored with EC. $D_3$ conveyed the information related to bacterial contamination, regardless of the water chemical profile.

**Table 1.** Correlation matrix of the 15 physicochemical and biological parameters (5,295 observations).

| Param. | Enteroc. | E. Coli | EC | HCO$_3$ | H | Ca | Cl | Mg | K | Na | SO$_4$ | Fe | Mn | NO$_3$ | As |
|---|---|---|---|---|---|---|---|---|---|---|---|---|---|---|---|
| **Enteroc.** | 1 | | | | | | | | | | | | | | |
| **E. Coli** | **0.594** | 1 | | | | | | | | | | | | | |
| **EC** | −0.006 | −0.006 | 1 | | | | | | | | | | | | |
| **HCO$_3$** | 0.009 | 0.032 | **0.653** | 1 | | | | | | | | | | | |
| **H** | −0.016 | −0.013 | 0.221 | 0.116 | 1 | | | | | | | | | | |
| **Ca** | −0.007 | 0.002 | **0.861** | **0.766** | 0.150 | 1 | | | | | | | | | |
| **Cl** | −0.010 | −0.018 | **0.753** | 0.197 | 0.221 | 0.405 | 1 | | | | | | | | |
| **Mg** | 0.006 | 0.004 | **0.669** | 0.557 | 0.161 | 0.519 | 0.327 | 1 | | | | | | | |
| **K** | −0.011 | −0.008 | **0.617** | 0.293 | 0.165 | 0.448 | **0.604** | 0.339 | 1 | | | | | | |
| **Na** | −0.011 | −0.019 | **0.743** | 0.193 | 0.203 | 0.382 | **0.971** | 0.311 | **0.615** | 1 | | | | | |
| **SO$_4$** | −0.016 | −0.031 | **0.748** | 0.241 | 0.126 | **0.712** | 0.428 | **0.614** | 0.442 | 0.439 | 1 | | | | |
| **Fe** | −0.002 | −0.003 | 0.025 | 0.020 | 0.047 | 0.027 | 0.015 | 0.015 | 0.050 | 0.023 | 0.029 | 1 | | | |
| **Mn** | 0.006 | −0.011 | 0.143 | 0.048 | 0.144 | 0.050 | 0.183 | 0.073 | 0.157 | 0.221 | 0.083 | 0.114 | 1 | | |
| **NO$_3$** | 0.007 | 0.040 | 0.447 | 0.357 | 0.175 | 0.509 | 0.236 | 0.251 | 0.275 | 0.196 | 0.314 | 0.011 | −0.029 | 1 | |
| **As** | −0.006 | −0.011 | −0.107 | −0.169 | −0.022 | −0.14 | −0.023 | -0.079 | −0.031 | −0.014 | −0.036 | 0.030 | 0.030 | −0.045 | 1 |

**Table 2.** Eigenvalues and percentage of explained variance by the first principal components. Only PC$_1$ to PC$_6$ are taken into account in this study.

|  | PC$_1$ | PC$_2$ | PC$_3$ | PC$_4$ | PC$_5$ | PC$_6$ | PC$_7$ | PC$_8$ | PC$_9$ | PC$_{10}$ | PC$_{11}$ | PC$_{12}$ |
|---|---|---|---|---|---|---|---|---|---|---|---|---|
| Eigenvalue | 5.1 | 1.7 | 1.6 | 1.1 | 1 | 0.9 | 0.9 | 0.8 | 0.7 | 0.5 | 0.4 | 0.4 |
| Variance (%) | 34.1 | 11 | 10.4 | 7.3 | 6.7 | 6.3 | 5.8 | 5 | 4.3 | 3.2 | 2.9 | 2.7 |
| Cum. % | 34.1 | 45.1 | 55.5 | 62.8 | 69.4 | 75.7 | 81.5 | 86.5 | 90.9 | 94.1 | 96.9 | 99.6 |

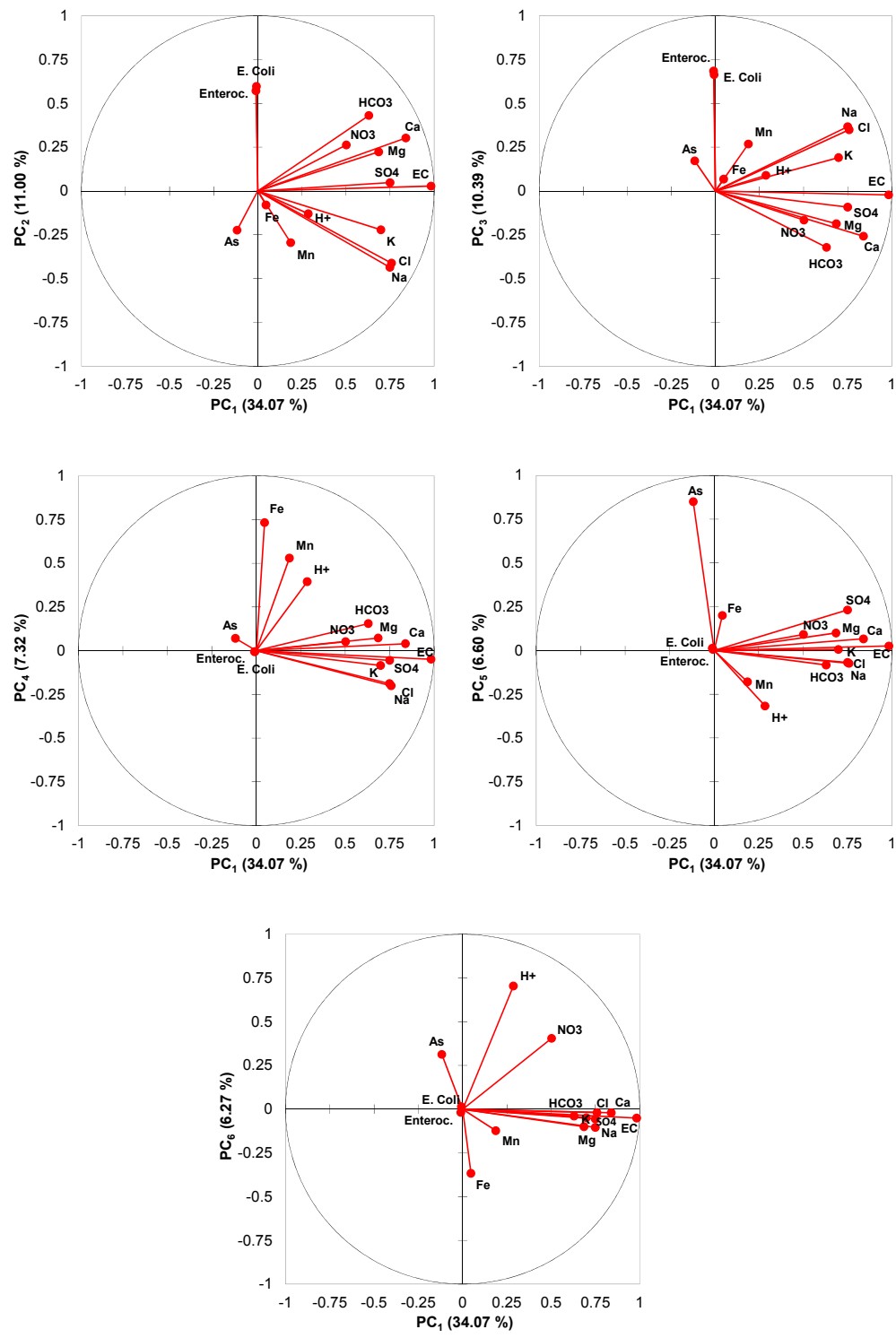

**Figure 3.** Distribution of the parameters on the six first PCs.

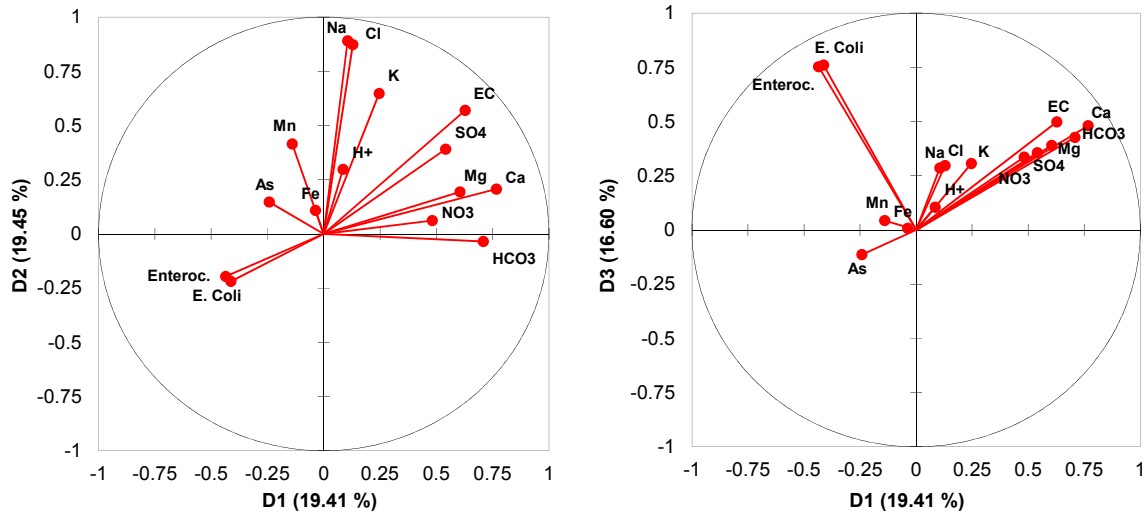

**Figure 4.** Distribution of the parameters on the PCs $D_1$, $D_2$ and $D_3$, after orthomax rotation.

### 3.2. Regional Distribution of the Principal Components

The distribution of $PC_1$ values is shown in Figure 5a. The lowest values (blue color) were observed in mountain areas with crystalline rock lithology. Slightly higher values (green color) corresponded to the alluvial valleys, such as, for example, the Durance valley. The areas with karst lithology had higher values and appeared in yellow. Finally, the highest values (orange and red color) are located in the Argens basin south of the PACA region and in coastal areas. It should be noted that the Rhone delta occupied by the plain of Camargue appears in blue on the map. This is actually an artifact because of, on the one hand, the absence of sampling points in this area, and on the other hand, the extrapolation of some wells with waters strongly influenced by the diluted waters of the Rhone River. The distribution of $PC_1$ values is broadly similar to that of EC, Mg, Ca, and $HCO_3$ (Figure 6).

The high $PC_2$ values that corresponded to high carbonates, calcium, magnesium contents, associated with the bacteriological parameters, were distributed in local spots, and generally within calcareous and karstic areas. The same type of distribution was observed for $PC_3$ but more clearly close to the coast. Some spots had high values for both $PC_2$ and $PC_3$. $PC_4$ was essentially associated with two spots in the coastal alluvial plains east of Toulon, at the foot of the Massif des Maures. The fifth factorial axis is weak and it carries only information related to a single physicochemical parameter, namely arsenic. The positive values of $PC_5$ were located mainly in mountain areas, and often associated with the border of the crystalline zones. To this distribution, we must add some local occurrences in the plains. High $PC_6$ values were limited to a few wells around the St Tropez and Frejus Gulfs surrounded by the Maures and Esterel crystalline massifs and one well in the Alps on the left bank of the Tinée River, an affluent of the Var crossing the crystalline Mercantour Massif.

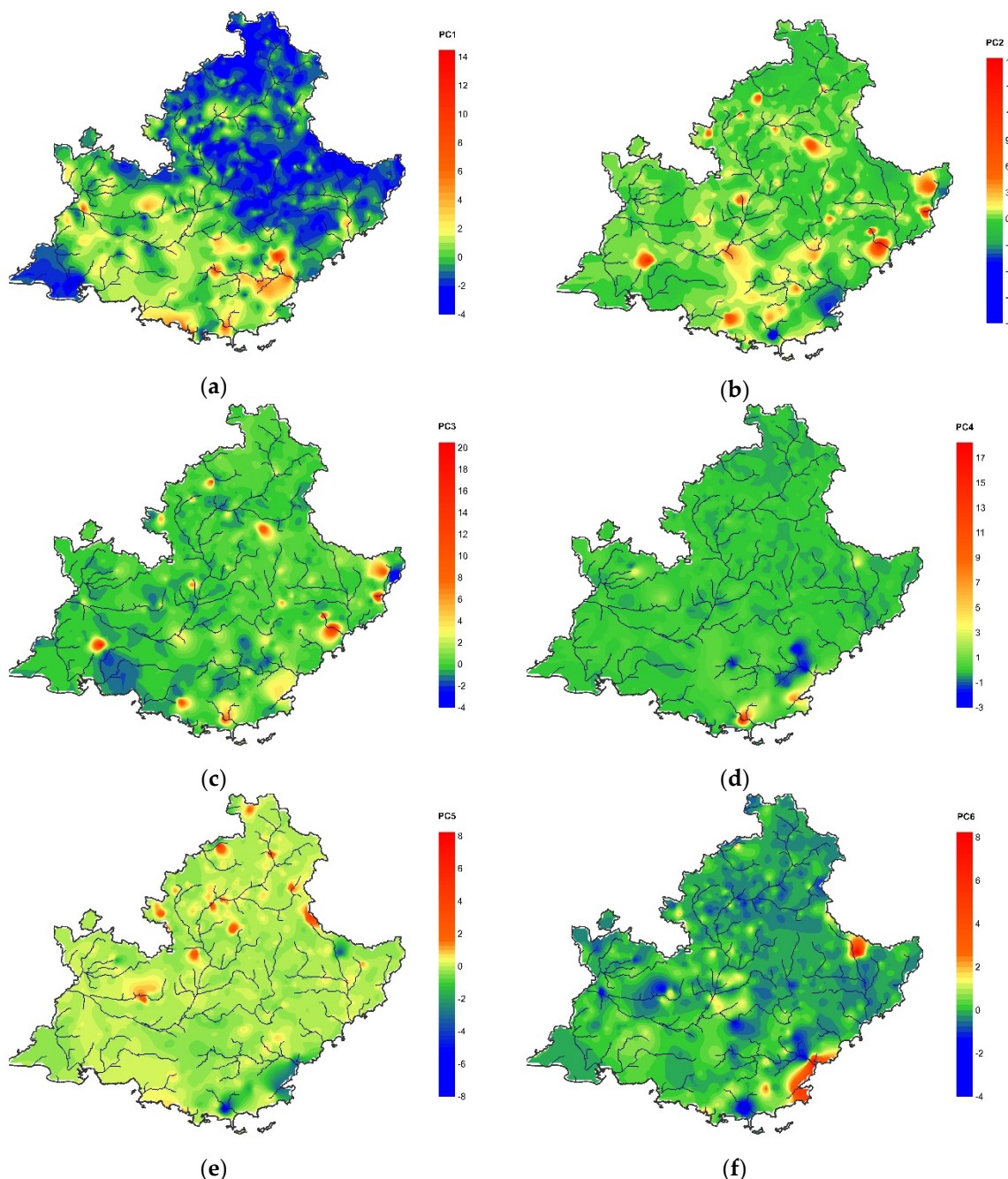

**Figure 5.** Distribution of the six first principal components (**a**–**f**, respectively) in the PACA Region.

## 4. Discussion

### 4.1. Dimension Reduction of the Hyperspace

The variance is significantly distributed over a large number of factorial axes. From $PC_2$ to $PC_4$, the percentage of explained variance is almost similar, around 10%, suggesting that the dataset represents a complex environment, where the parameters are not linked by obvious relationships. This is explained in particular by the great diversity of lithologies, altitudes, and land use at the scale of the studied area. In other words, there are many sources of variability of comparable importance, which justifies a posteriori the use of multivariate analysis methods for data processing, in order to identify, discriminate and rank the sources of variation in the water quality. The multiplication of the parameters enriches the database but complicates the data processing, increasing the number of dimensions and

the distances between the observations in the hyperspace. The dimensional reduction with the PCA reduces this disadvantage, combining redundant information into single principal components. The eigenvalue associated with the first factorial axis is 5.1, which means that this principal component alone convey, synthetically, more information than 5 original variables. In fact, its distribution over the study area (Figure 5a) presents the same trends as that of the EC values and of Ca, Mg, $HCO_3$ contents, etc (Figure 6). The transition from a 15- to a 6-dimensional hyperspace of data constitutes a strong reduction of the hyperspace of data, thus facilitating the information processing. The PCA treatment highlights sources of variability classically observed in hydrochemistry, in particular the water mineral load, the water-rock interactions, redox processes, and possible alterations often of anthropogenic origin.

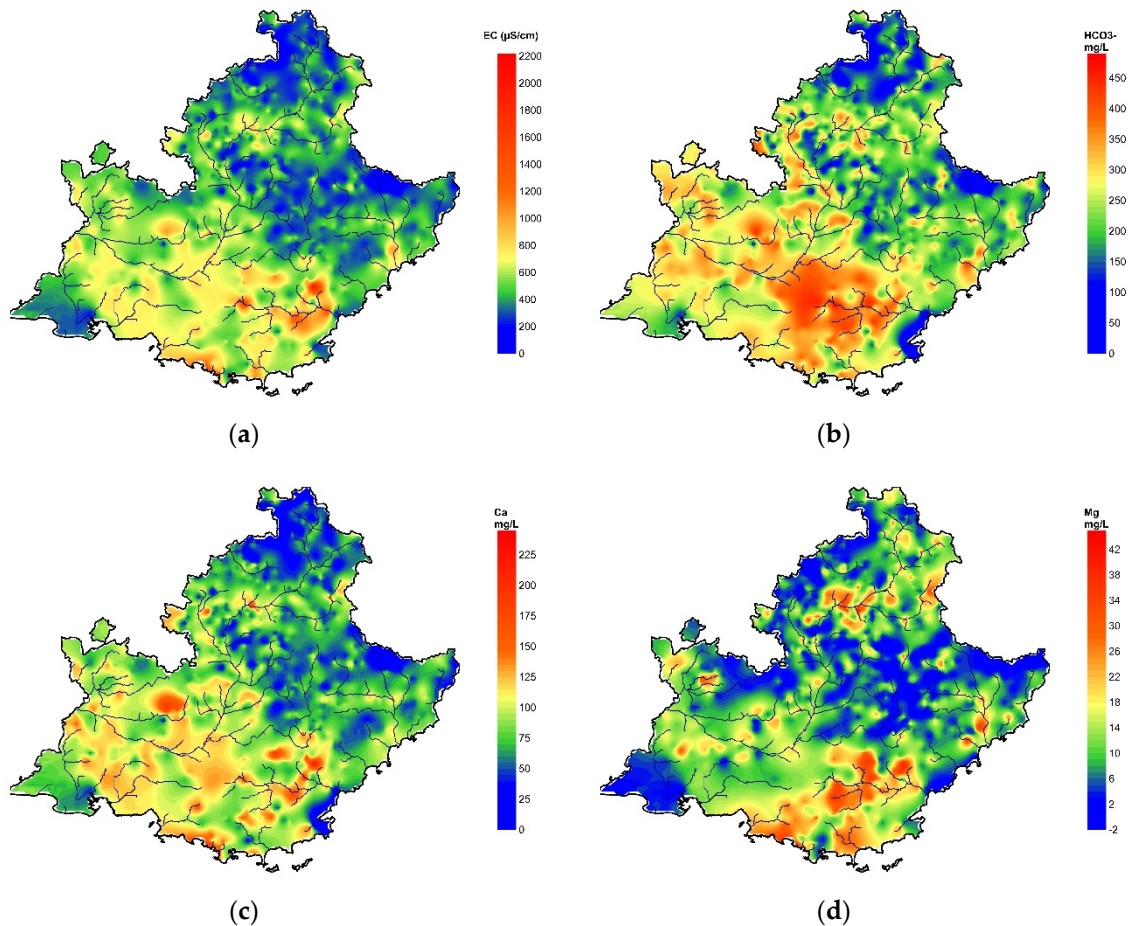

**Figure 6.** Distribution of the original variables (**a**) EC, (**b**) $HCO_3$, (**c**) Ca and (**d**) Mg on the PACA region. Note a rather similar distribution of these parameters, which conveys redundant information. This redundancy is concentrated on the first principal component of the PCA (Figure 5a).

## 4.2. Water-Rock Interaction and Coastal Marine Intrusions

Both the chemical profile and the regional distribution of the PC values make it possible to follow the water path and the acquisition and/or changes in chemical characteristics in detail. The first principal component reflected the contrast between, on the one hand, the low mineralized waters of high mountain areas or crystalline massifs, and on the other hand, the more mineralized waters occurring close to the Mediterranean coast. In high mountains, water transit times are reduced due to the steep slopes. The prevalence of crystalline rocks and colder weather conditions reduce weathering rates. Consequently, the groundwaters captured in these sectors show low values on $PC_1$.

The frequent presence of black pyritous marls on the periphery of the crystalline massifs alter the water chemical profile, mainly by a moderate increase in sulfate contents. Due to the presence of pyrite, these waters show low redox conditions, are slightly more acidic, and the Fe and Mn contents are slightly higher. These aquifers are often shallow and vulnerable to contamination, and therefore are positively scored on $PC_3$. When the waters arrive in the main valleys, the water-rock time of contact increase. Locally, the stagnation of low gradient groundwaters upstream natural rock bars logically results in an increase in the coordinates on $PC_1$, as it can be observed along the course of the Durance, from the upper basin to its confluence with the Rhone River. The higher solubility of carbonate rocks contributes to an increase in electrical conductivity, which varies according to the seasons but also according to more or less dry multi-annual periods [23]. In the south of the PACA region, the Argens basin intersects a Triassic formation rich in much more soluble evaporites, which impose a $Ca-Na/SO_4$ chemical profile to the water, and contribute to much higher values of $PC_1$. Finally, marine intrusions in coastal alluvial aquifers are particularly visible around the cities of Toulon and Nice, leading to higher EC and high water coordinates on $PC_1$. The saline bevel progress in the continent because of the freshwater pumping. The orthomax rotation (Figures 4 and 7) makes it possible to better distinguish two main chemical profiles, namely the influence of calcareous rocks on $D_1$ axis, and marine intrusions on $D_2$ axis. Sulfate rich waters are logically in an intermediate position.

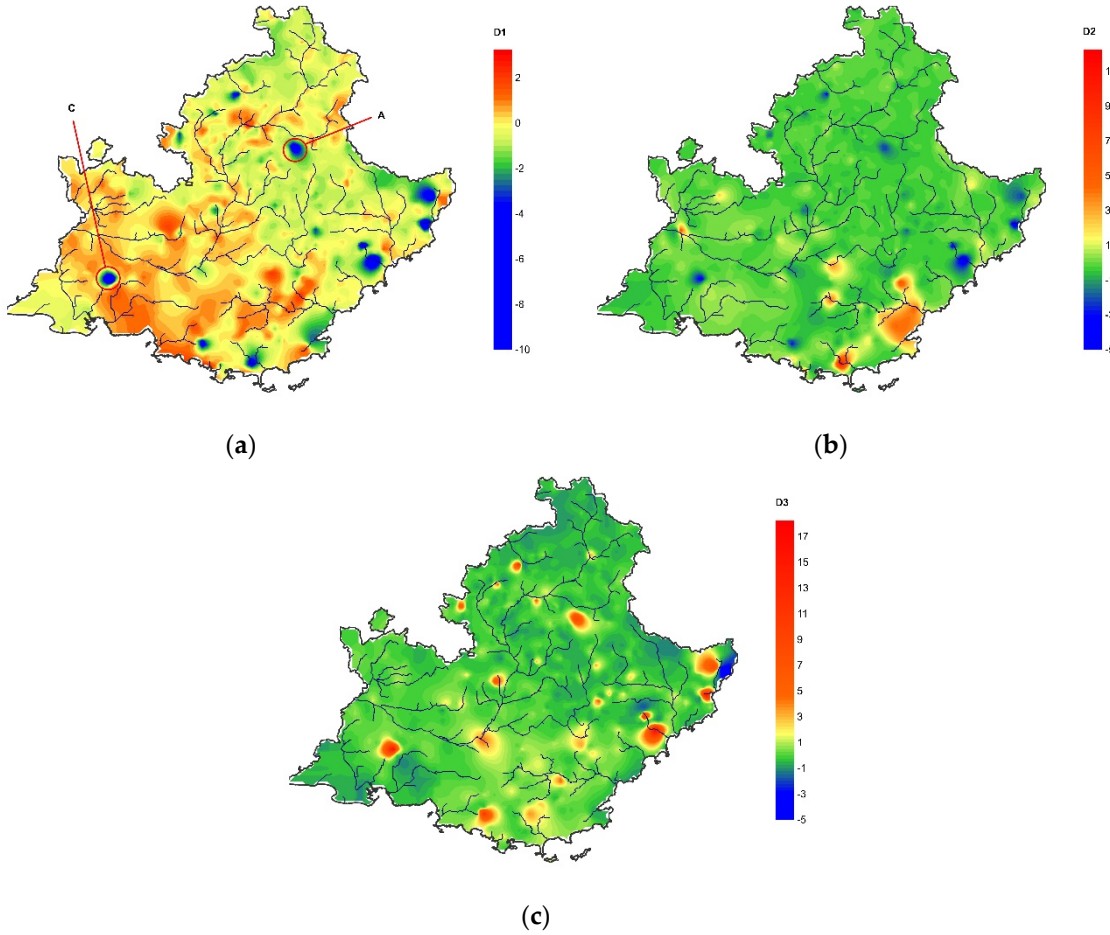

**Figure 7.** Distribution of the three first principal components D1, D2 and D3 (**a**, **b** and **c**, respectively) after orthomax rotation.

*4.3. Discrimination of Bacteriological Contamination*

The bacteriological parameters are mainly expressed in the 2nd and 3rd principal components. Such water fecal contamination can be linked to anthropogenic sources, but especially to sheep and

cattle animal droppings, as in grazing areas that cover most of the mountain and mid-altitude areas. Considering the surface of the study area, there is a diversity of cases, depending on the domestic or animal origin of the contamination, including the seasonal tourist impact more marked towards the coast, and the seasonal migration of herds to the higher mountain pastures. This diversity makes this parameter not correlated to other sources and appears as an independent source of variability.

The second factorial axis has higher values in calcic and/or karstic areas (Figure 5b), which are characterized by fast flows, both from surface water towards the water table, and within the water tables. These rapid flows associated with significant turbidity are known factors of bacteriological vulnerability for these aquifers [27–30]. The contamination, however, depends on the local context, namely the depth of the aquifer but also the presence or absence of large supply conduits of the saturated area, of breeding close to the catchment area, of wastewater treatment plants. For all these reasons, the vulnerability is heterogeneous, the bacterial contaminations can occur locally and high $PC_2$ values appear per spot. Bacteriological contamination is also carried by $PC_3$, associated with a Na/Cl type chemical profile. It reflects the strong demographic pressure (increasing in summer) in coastal areas, with shallow aquifers and weak flows, increasing their vulnerability to contamination. The absence of livestock farming in these coastal areas is one of the factors that explains why nitrates are negatively scored in $PC_3$.

To summarize, both $PC_2$ and $PC_3$ convey information related to microbiological contamination of water. $PC_2$ relates to calcareous and karst areas, whereas with $PC_3$ we find other contaminated environments, and in particular the coastal plains. The distribution of bacteriological parameters on two factorial axes underlines the fact that the contamination processes are complex. After orthomax rotation, $D_3$ includes all the contexts where bacterial contamination occur in PACA region. Both axes $D_1$ and $D_2$ are negatively correlated with the bacteriological parameters. Some specific spots, for example near the city of Salon de Provence or in the upper Durance watershed (respectively C and A in Figure 7a), with negative values in $D_1$ and $D_2$ correspond to water resources close to the surface and prone to contamination. C is due to local infiltration into the Crau aquifer, supplied by the diversions from the Durance canal, whereas A is a catchment in the Annot sandstone, a porous material with fast flow and high turbidity.

### 4.4. Redox Conditions

The fourth PC, with contrast parameters such as $NO_3$ with Mn and Fe, reflects the oxidation-reduction state of water. The most reductive waters had negative values; anoxic waters showed close to 0 or slightly positive values, and oxidizing waters had positive values. In aerated waters, denitrification is not active. When contaminated, these waters have high levels of nitrates, while oxygenation and the more alkaline conditions limit the levels of dissolved Fe and Mn. In contrast, the more anoxic and reducing waters, in which denitrification limits the levels of $NO_3$, favor the presence of soluble Fe and Mn in the reduced state. The regional distribution of $PC_4$ (Figure 5d) emphasizes that the waters of karstic or crystalline and steep-sloped areas may have oxidative conditions, whereas alluvial aquifers in low-lying and low-slope areas with lower flow are more likely to have anoxic conditions. Such anoxic waters are observed in the accompanying aquifers of the Rhone and the lower Durance valley, as well as in the Sorgues plain close to the city of Avignon. Aquifers located in sulfide containing black marl of the Cretaceous also have pronounced reducing characteristics.

Around the crystalline massifs of Mercantour, the Maures and Esterel, the waters not influenced by carbonates have a rather low pH compared to the rest of the region. Steep slopes and torrential regime of rivers are associated with water tables circulating in environments made up of large blocks and coarse materials, highly aerated and suitable for the oxidation of ammonium to nitrates. These very local conditions are reflected on $PC_6$ (Figures 3 and 5f). Its weight over the total variance is low due to the small number of samples concerned at the scale of the PACA region.

The distribution of iron content (not shown) differs from the distribution of $PC_4$ values, which may seem paradoxical at first glance. This discrepancy is explained by the presence of iron in two

forms, on the one hand in reduced form soluble in anoxic aquifers, and on the other hand in the form of ferric iron colloids in the oxidizing waters of karstic aquifers with higher turbidity. These colloids being quantified during the analysis, high iron contents also accompany the karstic aquifers. Thus, the fourth factorial axis opposing nitrate to iron and manganese reflects the dynamics of these three elements according to the redox character of water.

### 4.5. Occurrence of As

Arsenic is present in trace amounts, generally of the order of 1 to 5 µg $L^{-1}$ both in surface waters and in aquifers [31], whereas WHO guidelines for drinking waters are fixed at 10 µg $L^{-1}$. It may be more abundant under certain highly alkaline conditions [32] but not occurring in the PACA region, or in certain geological contexts, and associated with sulfide minerals (As-pyrites). In the latter case, the presence of dissolved arsenic should be positively correlated with sulfates and $H^+$, and negatively with carbonate alkalinity. This is observed in the analysis of this dataset with extreme values that reach 76 µg $L^{-1}$. However, because As-metalliferous deposits are few in the PACA region, then the weight of this fifth PC in the total variance remains low (6.7%). The few higher values of $PC_5$ in the plain (Figure 5e) could be due to the burning of vine shoots formerly treated with arsenic-containing pesticide mixture, former practices that were frequent in vineyards [33,34].

### 4.6. The Limits of this Approach and the Future Directions

PCA is proved to be a powerful tool to reduce the dimension of the dataset, while losing a small part of the information contained. Consequently, the number of maps necessary to represent the diversity of the waters is reduced. Most of the information can be summed up in 6 instead of 15 parameter distribution maps, which is a major advantage. The identification of the processes responsible for the diversity of waters is facilitated, but the analysis of this dataset also highlights the limits of this approach. The correlation matrix (Table 1) is characterized by fairly low values between the variables. Processing the entire dataset brings together very different environments and mixes different sources of variability specific to each landscape unit or each groundwater body, which induces weak correlations between the parameters. Consequently, the importance of each factorial axis is averaged on a regional scale. This heterogeneity is mainly due to the diversity of the natural environments, altitudes and human activities of the study area, that can mark the quality of groundwater. The analysis could be greatly improved by distinguishing more homogeneous subsets within the PACA region, which could be based on landscape units, or groups of groundwater bodies presenting the same sources of variability in water quality.

### 5. Conclusions

The PACA region, in the southeast of France, has a great geological and topographical diversity together with seasonal human activities. Because of this diversity of the natural environment and its use, groundwater resources intended for human consumption are highly variable in terms of physicochemistry and bacteriological vulnerability. The SISE-Eaux database collected by the Regional Health Agency (ARS) is extremely rich in information, but its valuation is delicate because of (1) the strong redundancy of information conveyed by the various parameters of the database, and (2) the diversity of processes involved that may change each parameter.

The treatment through principal component analysis made it possible to overcome this redundancy, and to synthesize the information carried by different parameters into single macro-parameters. The dimensionality of the information contained in the database is thus reduced without significant loss of information. In the case treated, the initial 15D-hyperspace was reduced to 6D with a loss of the order of 25% of the information, including the statistical noise that is inherent in any database. This treatment also makes it possible to separate these independent sources of variability, and hence the information conveyed by the water composition. The reading and interpretation of information are facilitated and it becomes easy to draw synthetic maps of their distribution based on the principal

components. The effects of water mineral load, lithology, marine intrusions, redox conditions, arsenic occurrence and vulnerability to bacteriological contamination have been distinguished.

This work carried out at the scale of the PACA region (31,400 km$^2$) also shows the limits of this approach. At this regional scale, very different environments are grouped each with their own sources of variability. Such an approach leads to mixing all these sources of variability and diluting them throughout the dataset, making their identification more difficult. To overcome this disadvantage, a similar approach but at the scale of each groundwater body should allow to establish relevant typology of groundwaters based on better discriminate sources of variability and associated processes, thus providing relevant tools for the sustainable management of each groundwater bodies.

**Author Contributions:** Conceptualization, A.T. and S.Y.; methodology, A.T., S.Y., L.B. and V.V, formal analysis, A.T., L.B., T.B. (Tarik Bouramtane) and V.V.; investigation, A.T., S.Y. and V.V., resources, F.D., M.M. (Marc Moulin); data curation, A.T., S.Y., T.B. (Tarik Bouramtane) and V.V.; writing—original draft preparation, A.T. and L.B.; writing—review and editing, L.B.; visualization, A.T. and L.B.; supervision, I.K.; project administration, A.T. and I.K.; funding acquisition, T.B. (Tarik Bahaj) and M.M. (Moad Morarech) All authors have read and agreed to the published version of the manuscript.

**Funding:** This research received no external funding.

**Acknowledgments:** Authors are grateful to the Regional Health Agency (Agence Régional de la Santé, ARS-PACA) for the provision of the Size-Eaux database. James Hesson of Academic English Solutions (AcademicEnglishSolutions.com) proofread the manuscript.

**Conflicts of Interest:** The authors declare no conflict of interest. The funders had no role in the design of the study; in the collection, analyses, or interpretation of data; in the writing of the manuscript, or in the decision to publish the results.

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
