# Peer review of "Dimension Reduction and Analysis of a 10-Year Physicochemical and Biological Water Database Applied to Water Resources Intended for Human Consumption in the Provence-Alpes-Côte d’Azur Region, France"

_water, doi:10.3390/w12020525_

Round 1

Reviewer 1 Report

Dear Authors,

Your manuscript Dimension reduction and analysis of a 10-year physicochemical and biological water database: Applied to water resources intended for human consumption in the PACA region, France deals with very interesting subject. It is very nicely describe when you have huge data base how to ‘’quickly’’ get an overview about water quality in the regional scale. My opinion is that your manuscript should be publish in the journal after minor corrections which are:

Line 59 instead of ''…''put etc

Line 80 after Provence-Alpes-Côte d'Azur put in bracket the abbreviation name (PACA)

Line 92 please enlarge figures because they are so small and very difficult to read

Line 100 is it … represents etc or written by mistake. If it is in the meaning of etc write this please

Line 112 correct it as in line 92

Line 281 the same comment as in the line 100

Line 331 the same comment as in the line 100

Line 385 you are mention higher As concentration due to activities in vineyards, what about Cu in this part of you study area? Because Cu was used in fungicides.

Author Response

Thank you for this favorable evaluation for the publication of this manuscript. The answers are listed below.

Your manuscript Dimension reduction and analysis of a 10-year physicochemical and biological water database: Applied to water resources intended for human consumption in the PACA region, France deals with very interesting subject. It is very nicely describe when you have huge data base how to ‘’quickly’’ get an overview about water quality in the regional scale. My opinion is that your manuscript should be publish in the journal after minor corrections which are:

Line 59 instead of ''…''put etc OK, it has been changed

Line 80 after Provence-Alpes-Côte d'Azur put in bracket the abbreviation name (PACA). OK, inserted

Line 92 please enlarge figures because they are so small and very difficult to read OK, the size has been increased

Line 100 is it … represents etc or written by mistake. If it is in the meaning of etc write this please Ok, it has been modified

Line 112 correct it as in line 92. OK, the size has been increased similarly.

Line 281 the same comment as in the line 100. Ok, modified

Line 331 the same comment as in the line 100. Ok, modified

Line 385 you are mention higher As concentration due to activities in vineyards, what about Cu in this part of you study area? Because Cu was used in fungicides. Data about Cu contents in groundwater were too sparse and have not been retained for this study. Cu contents will, however, be incorporated in a future study that deals more specifically with metals.

Reviewer 2 Report

The aim of the paper entitled Dimension reduction and analysis of a 10-year physicochemical and biological water database : Application to water resources intended for human consumption of the PACA region, France is to present results of reducing the dimensionality of the SISE-Eaux database and to identify water-rock interactions, bacteriological contamination, redox processes and arsenic occurrence as the main sources of variability. Unfortunately the aim is not highlighted. Introduction part is too short. Different methods connected with data analysis should be described and and the choice of PCA should be justified. The PCA method was described correctly, as well as the test results. However, it makes sense to use more advanced research methods such as artificial neural networks to classify the results obtained.

Author Response

Thank you for this constructive evaluation for the publication of this manuscript. The answers are listed below.

Paper based on statistical treatment of an extensive water quality database by PCA (principal component) analysis. Based on interpreted data, authors come to deductions on groundwater quality origin, what can be perceived as inverse task in classical regional hydrogeochemical surveys. Nevertheless, such an approach can be spatially expressed in maps based on numeric values, what is perhaps more sophisticated than traditional regional hydrogeochemical report.

Some more recommendations can be found here:

Paper name:

- in the paper name, “PACA” region should be given in its full name (Provence-Alpes-Côte d’Azur region), as for those not living in France it can be perhaps understood as e.g. “Paris – Atlantic Calcareous Aquifer” OK, it has been mentioned in the title text.

Abstract:

- describes the paper content in the desired format OK

 Key words:

- keywords (hydrochemistry, water resource, hydrogeology, multivariate statistics, France) are adequately selected, “PACA region” should be omitted or explained here in whole; Ok, PACA Region has been removed

Text:

- Figure 1: the region should be introduced on more general map (Europe / France / Region); OK, a map of France with the location of the PACA region has been added.

- lines 93-100: description is too general and perhaps slightly unprofessional, it should be given more quantitatively or the space should be given to more adequate citations; We agree. This part has been rewritten based on the chronological events in the local geology, with additional references.

- line 139: formulation “occupation of the grounds” should be perhaps given as  “occupancy of the grounds”; It has been simplified using “land use”

- lines 341 – 343: Two consecutive sentences begin with “After orthomax rotation”. Indeed, it has been corrected

- lines 383 – 385: it should be more clearly explained how higher values of PC5 can be influenced by burning wine shoots and treated wooden trellises in wineyards! The relationship between these former practices (using As containing pesticides) and PC5 is now better explained.

Reviewer 3 Report

Paper based on statistical treatment of an extensive water quality database by PCA (principal component) analysis. Based on interpreted data, authors come to deductions on groundwater quality origin, what can be perceived as inverse task in classical regional hydrogeochemical surveys. Nevertheless, such an approach can be spatially expressed in maps based on numeric values, what is perhaps more sophisticated than traditional regional hydrogeochemical report.

Some more recommendations can be found here:

Paper name:

- in the paper name, “PACA” region should be given in its full name (Provence-Alpes-Côte d’Azur region), as for those not living in France it can be perhaps understood as e.g. “Paris – Atlantic Calcareous Aquifer”

Abstract:

- describes the paper content in the desired format

 Key words:

- keywords (hydrochemistry, water resource, hydrogeology, multivariate statistics, France) are adequately selected, “PACA region” should be omitted or explained here in whole;

Text:

- Figure 1: the region should be introduced on more general map (Europe / France / Region);

- lines 93-100: description is too general and perhaps slightly unprofessional, it should be given more quantitatively or the space should be given to more adequate citations;

- line 139: formulation “occupation of the grounds” should be perhaps given as  “occupancy of the grounds”;

- lines 341 – 343: Two consecutive sentences begin with “After orthomax rotation”

- lines 383 – 385: it should be more clearly explained how higher values of PC5 can be influenced by burning wine shoots and treated wooden trellises in wineyards!

Author Response

The aim of the paper entitled Dimension reduction and analysis of a 10-year physicochemical and biological water database : Application to water resources intended for human consumption of the PACA region, France is to present results of reducing the dimensionality of the SISE-Eaux database and to identify water-rock interactions, bacteriological contamination, redox processes and arsenic occurrence as the main sources of variability. Unfortunately the aim is not highlighted. Introduction part is too short. Different methods connected with data analysis should be described and the choice of PCA should be justified. The PCA method was described correctly, as well as the test results. However, it makes sense to use more advanced research methods such as artificial neural networks to classify the results obtained.

Thank you for this constructive comments. The aim has been highlighted as suggested, specifying that the method aim to identify and map the main sources of variability. The introduction is modified as also requested by other referees. We agree that the introduction remained short, partly because the regional setting are incorporated as 2.1. in the Materials and methods section.

We agree with the comment. However, in this paper, we used only classical linear statistics for data treatment, but further ongoing researches compare these classical methods with others non-linear machine learning methods.